# High-Performance Detection-Based Tracker for Multiple Object Tracking in UAVs

**Xi Li [1], Ruixiang Zhu [1], Xianguo Yu [2],\* and Xiangke Wang [2]**

[1] Hunan Provincial Key Laboratory of Flexible Electronic Materials Genome Engineering, Changsha University of Science and Technology, Changsha 410004, China
[2] College of Intelligence Science and Technology, National University of Defense Technology, Changsha 410073, China
\* Correspondence: yuxianguo11@nudt.edu.cn

**Abstract:** As a result of increasing urbanization, traffic monitoring in cities has become a challenging task. The use of Unmanned Aerial Vehicles (UAVs) provides an attractive solution to this problem. Multi-Object Tracking (MOT) for UAVs is a key technology to fulfill this task. Traditional detection-based-tracking (DBT) methods begin by employing an object detector to retrieve targets in each image and then track them based on a matching algorithm. Recently, the popular multi-task learning methods have been dominating this area, since they can detect targets and extract Re-Identification (Re-ID) features in a computationally efficient way. However, the detection task and the tracking task have conflicting requirements on image features, leading to the poor performance of the joint learning model compared to separate detection and tracking methods. The problem is more severe when it comes to UAV images due to the presence of irregular motion of a large number of small targets. In this paper, we propose using a balanced Joint Detection and Re-ID learning (JDR) network to address the MOT problem in UAV vision. To better handle the non-uniform motion of objects in UAV videos, the Set-Membership Filter is applied, which describes object state as a bounded set. An appearance-matching cascade is then proposed based on the target state set. Furthermore, a Motion-Mutation module is designed to address the challenges posed by the abrupt motion of UAV. Extensive experiments on the VisDrone2019-MOT dataset certify that our proposed model, referred to as SMFMOT, outperforms the state-of-the-art models by a wide margin and achieves superior performance in the MOT tasks in UAV videos.

**Keywords:** unmanned aerial vehicles; multi-object tracking; tracking by detection; set-membership filter





## 1. Introduction

Multi-Object Tracking (MOT) has emerged as a crucial technology in the field of computer vision [1–5], offering significant potential for a wide range of applications, including intelligent driving, behavioral cognition [6], and advanced video analysis. The primary objective of MOT is to analyze and interpret the trajectories of targets in video footage. Recently, Unmanned Aerial Vehicles (UAVs) have gained significant popularity due to their versatility and convenience, resulting in a surge of research interest in MOT for UAV applications.

Detection-based tracking (DBT) is the most widely adopted paradigm in MOT [3,5,7], including in the context of UAV-related MOT. This paradigm typically involves obtaining potential boxes and appearance information for each object in each frame, followed by applying a matching algorithm based on motion cues [8] and appearance information [2] to associate objects across adjacent frames. The acquisition of detection results and appearance information is commonly accomplished through neural networks. The Joint Detection and Re-Identification (JDR) model has recently attracted greater interest for obtaining both detection results and appearance information from images, particularly with the

development of multi-task learning. The JDR model involves training a single network for both detection and re-identification (Re-ID) tasks, aiming to balance performance and efficiency.

Despite the potential benefits of joint learning for detection and tracking tasks, the conflicting requirements of both tasks on image features can result in a poorer performance compared to separate detection and tracking methods. In the context of UAV-related MOT, the presence of numerous small objects and irregular object movement, compounded by the motion of the UAV itself, also pose significant challenges for MOT models. Additionally, as shown in Figure 1, the sudden and rapid movements of UAVs during flight can result in the extreme scenes in UAV views (this does not include extreme scenes caused by natural conditions, such as nighttime, storms, blizzards, etc.), thereby exacerbating the challenges faced in matching.

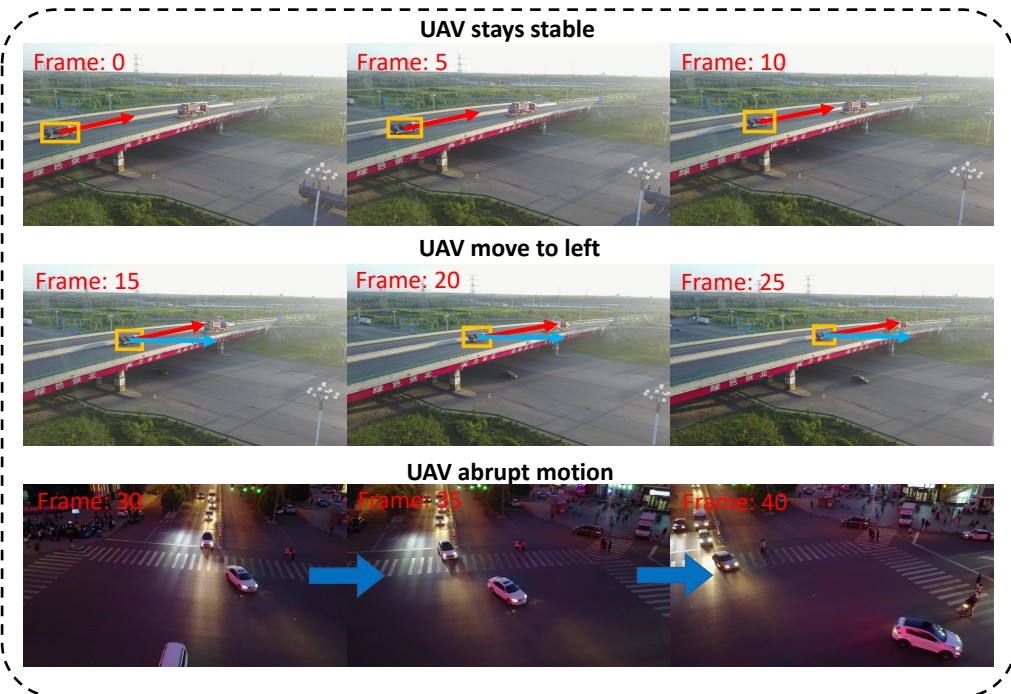

**Figure 1.** Irregular motion of objects in UAV video sequences. From frame 0 to frame 10, the UAV is stationary and the object is moving in the direction of the red arrow. From frame 15 to frame 25, the UAV is moving to the left while the object is moving in the direction of both the red and blue arrows in the image. From frame 30 to frame 40, the UAV is moving rapidly from left to right, resulting in significant changes in the state of object.

This study presents SMFMOT, a high-performance MOT tracker designed specifically for UAV videos. The proposed SMFMOT approach includes a symmetric shunt architecture-based anchor-based JDR network, which balances the detection and Re-ID tasks. By leveraging the benefits of dense anchor boxes to enhance model targeting recall [9], the JDR model built on top of an anchor-based detector is capable of effectively handling UAV video scenes with numerous small objects. Notably, we address the challenge of the lack of correspondence between anchors and Re-ID features in anchor-based JDR networks by developing a Re-ID feature learning module for each anchor in our design.

To improve the handling of irregular coupled movement of objects in drone video footage during the matching algorithm, we utilize the Set-Membership filter (SMF) [10] to predict and update the state of trajectories, in which the state is described by a bounded set. Based on this characteristic of SMF, we introduce an appearance-matching cascade (AMC) module that enhances matching accuracy by selecting objects within the prior range of the trajectory. We find that this module effectively addresses matching errors caused by similar appearances of different targets, as it utilizes strict screening of candidates based on state

information. Additionally, we propose a motion-mutation filter (MMF) module to address the issue of abrupt motion in UAV, which incorporates diverse matching strategies based on the UAV motion state.

We conduct experiments to assess the performance of SMFMOT on the benchmark dataset Visdrone2019-MOT [11]. The results of our experiments demonstrate that SMFMOT outperforms previous state-of-the-art methods. The motivation for this study stems from the unique characteristics of object motion in unmanned aerial vehicle video sequences. In summary, our contributions are as follows.

1. We propose a balanced JDR network for MOT, which is achieved using a symmetric framework design that ensures fairness between the detection task and Re-ID task.
2. To address the issue of irregular object motion, we utilize the SMF for trajectory state prediction and update, in which the state is described by a bounded set. Based on the bounded set, we design the AMC module to accurately select candidates for appearance matching, thereby reducing false matches.
3. We propose a MMF module to address the abrupt movement of the UAV, which determines the matching strategy based on the UAV motion state.

## 2. Related Work

Detection-based tracking approaches, as evidenced by methods such as SORT, Deep-SORT, UAVMOT, ByteTrack, JDE, and FairMOT [1–3,5,12,13], are currently the most effective methodologies for MOT. These approaches involve detecting targets in each frame and associating them via a matching method. Accordingly, this section primarily discusses recent advancements in multi-object detectors, Re-ID methods, matching methods, and motion models used in matching methods.

### 2.1. Detection Method

Object detectors can be divided into two main categories: anchor-free and anchor-based. The anchor-based detector regresses objects using anchors that are preset according to certain rules. For instance, YOLOv1 [14] divides the image into multiple sub-grids and generates a specific number of anchors in each sub-grid. It adopts a sample-to-sample matching strategy to obtain the prediction box. Similarly, Faster R-CNN [9] uses the regional proposal network (RPN) to filter out anchors generated on the feature map to achieve detection. The anchor-based detector is a powerful method for identifying small objects in UAV views, as the use of anchors improves the ability of the model to target recall.

Anchor-free object detectors perform object classification and localization directly on the image without the use of anchors. For example, CornerNet [15] adopts the Keypoint-based method to detect objects by predicting the upper-left and lower-right coordinates of the object surrounding box. Similarly, CenterNet [16] outputs the central points of targets through a heat map, which enables the network to regress the category, position, and size of the target through these center points.

### 2.2. Re-ID Method

Re-ID aims to obtain the appearance characteristics of objects as a vector, which is used to calculate the similarity of objects between frames to complete the matching. For example, JDE [13] is designed on top of YOLOv3 and obtains Re-ID features while locating the target. MOTS [17] is built by adding a Re-ID head on top of Mask R-CNN [18], which achieves high inference speeds.

### 2.3. Matching Method

Data association is the most crucial aspect of MOT, particularly in the context of detection-based tracking approaches, which rely on matching methods to associate objects across adjacent frames and assign the same ID to the same object. In general, matching algorithms mainly rely on the appearance information and motion patterns of objects.

### 2.3.1. Matched Based on Appearance Information

Recent work [2,3,5,13,17] has attempted to leverage appearance information for object association in MOT. These methods typically employ Re-ID networks to extract physical features of targets from object areas of the image and then calculate the similarity between detections and trajectories. Other works, such as TTU [19], and MPT [20], have focused on enhancing appearance features to improve the reliability of matching. For instance, TTU [19] has devised an online learning method to cope with appearance transformation, while MPT [20] has attempted to enhance appearance features by learning body postures. However, in scenarios in which a large number of similar objects are present in a single frame, relying solely on appearance information may become less reliable.

### 2.3.2. Matched Based on Motion Sign

The SORT algorithm is a classic method in MOT, which involves detecting objects in each frame followed by data association between objects and trajectories using the Kalman filter [21] and Hungarian algorithm. The IOU-Tracker [22] calculates IOU costs between trajectories and objects for association using the Hungarian algorithm, without utilizing motion information. These methods have been widely implemented due to their high efficiency and simplicity. However, both methods perform poorly in dense scenes, camera views with fast-moving objects, or scenes with objects in fast motion. As such, matching methods based solely on motion information may struggle to complete MOT tasks in extreme scenarios.

### 2.3.3. Matched Based on Appearance Information and Motion Sign

Recently, multiple studies have aimed to enhance the performance of matching methods in MOT by incorporating both motion models and appearance information, including DeepSORT [2], which utilizes a cascade matching process and matching by appearance similarity to improve MOT performance, and MOTDT [23], which employs a hierarchical data association strategy to address situations in which appearance information is unreliable and utilizes IOU matching when the appearance fails. Furthermore, UAVMOT [5] has developed an adaptive motion filter to improve MOT performance in UAV views. Although these methods commonly use Kalman filters to predict the prior state of the trajectory, which is assumed to conform to a Gaussian distribution, the motion of objects in the image is often coupled with the motion of UAV, potentially rendering the use of a Gaussian distribution with a mean of 0 to describe the error in state representation unreliable. Specially, MOTCM [24] adaptively adjusted the appearance matching window in order to address the issues caused by motion estimation errors. However, they only considered variations in position and size, while neglecting the fact that changes in velocity can also lead to matching errors.

### 2.4. Motion Model

The Kalman filter [21] is a widely used algorithm for state estimation and prediction in various scientific and engineering domains. Specifically, in MOT, it is utilized to predict the positions, sizes, and velocities of trajectories and match them with the corresponding objects. While the existing tracking algorithms, such as SORT and DeepSORT, assume the state of the trajectory to follow a Gaussian distribution with the mean as the state, the trajectory should be bounded within a range rather than a single value.

The SMF [10] is a nonlinear filtering algorithm that predicts the system state by constraining it within a prior range represented by a zonotope [25], while processing measurements. With the capability to perform state estimation with both Gaussian and non-Gaussian measurement noise, SMF is suitable for a wide range of applications. In this study, we utilize SMF to predict and update possible trajectory states and match the detection with the trajectory by verifying if it falls within the predicted set.

## 3. Methodology

### 3.1. Overall Framework

Our proposed model consists of two primary components. The first component is the balanced anchor-based JDR network, which extracts appearance feature vectors of detected objects during the detection process. The second component is the matching method, which builds upon DeepSORT and includes a SMF module, an AMC module, and an MMF module. Given a UAV video sequence $\left\{I_t \in \mathbb{R}^{W \times H \times 3}\right\}_{t=1}^{T}$ in which each frame is of size $W \times H$, our model aims to associate the same objects across different frames and assigns each trajectory a unique ID. The overall framework of our model is illustrated in Figure 2. The network takes two adjacent frames $I_t$ and $I_{t+1}$ as input. Then, the detection head outputs the categories $\{C\}_{i=1}^{N}$ and boxes $\{B\}_{i=1}^{N}$ of $N$ objects, while the Re-ID head outputs the appearance vectors $\{ID\}_{i=1}^{N}$ corresponding to objects. Object states are modeled and predicted using the proposed SMF module.

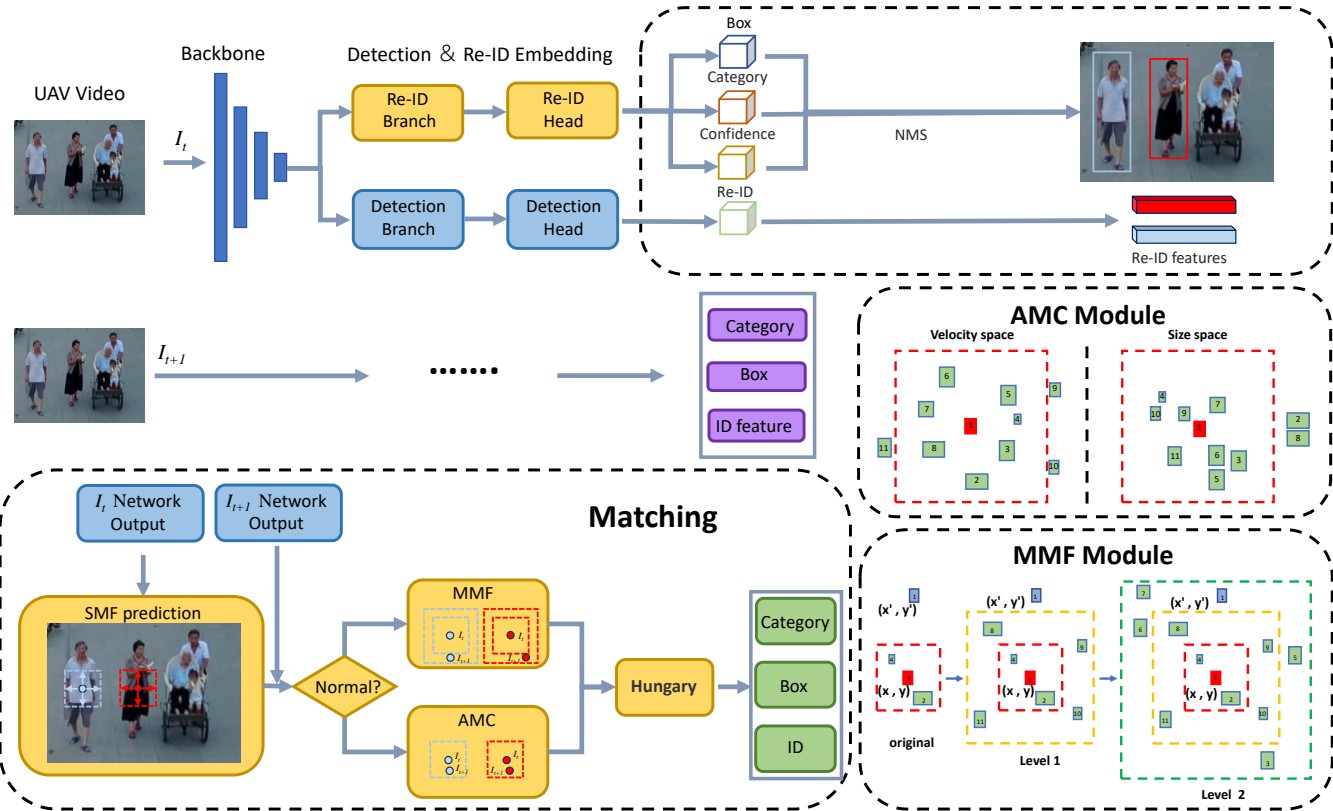

**Figure 2.** Overview of the proposed model. We propose a JDR network to retrieve object categories, box positions and appearance feature vectors from a pair of input images $I_t$ and $I_{t+1}$. In the tracking stage, we design the SMF module to predict trajectory states, the AMC module to select correct candidates for appearance matching, and the MMF module to tackle the challenge of rapid motion of UAVs. In AMC module and MMF module, the small rectangular patches with numerical values represent different objects; the red ones are objects in the last frames, the green ones and the blue ones are candidates in the current frame. The numbers in these rectangular patches represent the ID numbers of the objects.

### 3.2. JDR Network

To address the problem of simultaneous object detection and Re-ID, we propose an anchor-based JDR model consisting of a backbone, a detection branch, and a Re-ID branch. The network overview is presented at the top of Figure 2, in which we adopt the YOLOv7 backbone network to balance the performance and efficiency. An enhanced version of

the ELAN module, namely E-ELAN, is utilized in the backbone network, which employs shuffling, merging cardinality, and expansion to enhance the learning capability of the model while preserving the initial gradient path. This enables us to achieve high-quality Re-ID features and an excellent detection performance throughout the network.

Our detection branch is built on top of YOLOV7 backbone. The neck network converts and fuses the feature maps extracted from the backbone and outputs them to three output heads for object detection at different scales. Specifically, each output head is composed of three sub-heads, namely the box head, category head, and confidence head. The box head outputs both the location and size of the target, while the category head assigns each detected object to its corresponding category. Moreover, the confidence head determines the probability of the anchor containing the correct target, which serves as a positive sample.

To balance the performance between detection and Re-ID tasks, the Re-ID component is built as symmetric with the detection architecture, which is also equipped with three output heads that correspond to the appearance features of objects at different scales. In the Re-ID component, we learn the corresponding appearance features for each anchor in the detection component and train the Re-ID network as a classification assignment, in which the same object is considered to be in the same category. The appearance features vector for each object is mapped to a probability distribution vector $P = \{p(m), m \in [1, M]\}$ through a fully connected layer and softmax function. Denoting the ground truth ID numbers in one-hot representation as $T^i(m)$, we compute the Re-ID loss as follows.

$$L_{id} = -\sum_{i=1}^{N}\sum_{m=1}^{M} T^i(m) \log p(m) \tag{1}$$

where $N$ is the number of sequences and $M$ is the max number of ID in one sequence.

### 3.3. Matching Model

#### 3.3.1. Motion Module

In the proposed model, we employ the SMF [10] as the motion model to predict and update the state of the trajectory. For each trajectory, our tracking scene is defined in an eight-dimensional space $(u, v, \gamma, h, \dot{x}, \dot{y}, \dot{\gamma}, \dot{h})$, which includes the central point position $(u, v)$ of the object, the aspect ratio $r$, the length $h$ of the object box and their corresponding velocities $(\dot{x}, \dot{y}, \dot{\gamma}, \dot{h})$. We set $\mathbf{x}$ to be the range of each trajectory in the eight-dimensional space, and it can be spanned by a matrix $\mathbf{G}$ columns and centered by $\mathbf{c}$:

$$\exists(\mathbf{G}, \mathbf{c}) \in \mathbb{R}^{n \times n} \times \mathbb{R}^n : \mathbf{x} = \{\mathbf{G}\xi + \mathbf{c} : \|\xi\|_\infty \leq 1\} \tag{2}$$

The motion of the trajectory is modeled as a linear system, in which the noise is independent of the initial state. It can be described as follows:

$$\mathbf{x}_{k+1} = A\mathbf{x}_k + B\mathbf{w}_k \tag{3}$$

$$\mathbf{y}_k = C\mathbf{x}_k + D\mathbf{v}_k \tag{4}$$

where the state transition matrix is $A \in \mathbb{R}^{n \times n}$, the control matrix is $B \in \mathbb{R}^{n \times p}$, the measurement matrix is $C \in \mathbb{R}^{m \times n}$, and the feedforward matrix is $D \in \mathbb{R}^{m \times q}$. The system mandates that every trajectory adhere to the prescribed sequence of steps outlined below:

- **Initialization.** Set the initial prior range $[\![\mathbf{x}_0]\!]$.
- **Prediction.** For $k \in \mathbb{Z}_+$, the prior range is

$$[\![\mathbf{x}_k|y_{0:k-1}]\!] = A[\![\mathbf{x}_{k-1}|y_{0:k-1}]\!] \oplus B[\![\mathbf{w}_{k-1}]\!], \tag{5}$$

where $\oplus$ stands for the Minkowski sum and $y_{0:k-1}$ stands for $\{y_0, \cdots, y_{k-1}\} \in \mathbf{y}_k$. The prior ranges will be used for matching with the objects.

- **Update.** For $k \in \mathbb{N}_0$, given $y_k \in [\![\mathbf{y}_k]\!]$, the posterior range is

$$[\![\mathbf{x}_k \big| y_{0:k}]\!] = \mathcal{X}_k(C, y_k, D[\![\mathbf{v}_k]\!]) \bigcap [\![\mathbf{x}_k \big| y_{0:k-1}]\!], \qquad (6)$$

here we define $[\![\mathbf{x}_0]\!] := [\![\mathbf{x}_0 \mid y_{0:-1}]\!]$ for consistency, and $\mathcal{X}_k(C, y_k, D[\![\mathbf{v}_k]\!]) = \{x_k : y_k = Cx_k + Dv_k, x_k \in [\![\mathbf{x}_k]\!], v_k \in [\![\mathbf{v}_k]\!]\}$. The update stage is performed after matching trajectories with objects; the prior range $[\![\mathbf{x}_k \mid y_{0:k-1}]\!]$ is assumed to be infinitely large.

### 3.3.2. Appearance Matching Cascade Module

Thanks to the SMF algorithm that describes the each object trajectory as a bounded set, the matching process is conducted in a precise manner based on our specially designed AMC module. Concretely, only the detections contained in the predicted state set are considered when calculating the appearance cost matrix and performing trajectory–object matching. In contrast, DeepSORT uses a threshold based on the distance between the center points of the trajectories and detections to determine the appearance matching candidates, which may result in mismatches. As shown in Figure 3, DeepSORT selects candidates for appearance matching by considering only the location within a circular region, while ignoring the boundaries in size and speed.

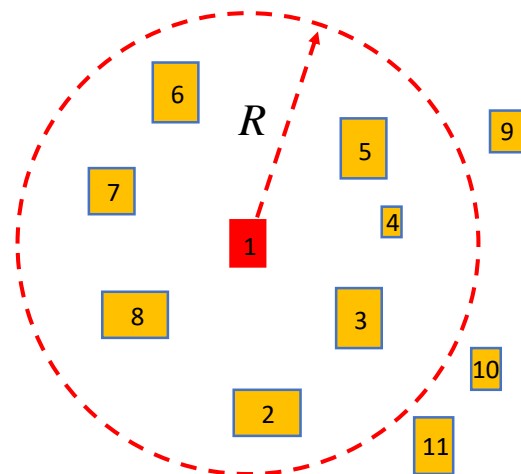

**Figure 3.** Overview of selecting candidates for appearance matching in DeepSORT. The small rectangular patches with numerical values represent different objects; the red one are objects in the last frames, the yellow ones are candidates in the current frame. The numbers in these rectangular patches represent the ID numbers of the objects. A positional threshold is used to select objects as the criterion for appearance matching. The candidates selected as appearance matches only need to have their center positions exist within a circular range of radius $R$ around trajectory 1.

In the AMC module, we only select objects whose position, size and corresponding velocity are contained within the trajectory prior range as candidates for appearance matching. As shown in Figure 2, there are seven appearance-matching candidates in the position space of trajectory 1, which is represented by a red box. However, some of these objects have aspect ratios that are not within the state space of trajectory 1, and therefore these objects should be excluded. Similarly, trajectories that are not within the other four velocity component spaces will also be excluded. Moreover, this module can alleviate the problem of misidentification between similar objects, as they frequently have diverse states. The basic process of overall AMC module is shown in Algorithm 1.

---

**Algorithm 1** AMC algorithm

---

**Input:** Track State $\mathcal{T} = \{track_1, \ldots, track_N\}_t^T$, $track_i$ is represented by Equation (4),
Detection $\mathcal{D} = \{det_1, \ldots, det_M\}_t^T$, $det_j = (u_j, v_j, \gamma_j, h_j, \dot{x}_j, \dot{y}_j, \dot{\gamma}_j, \dot{h}_j)$
**Output:** Matched trajectories and detections $\mathcal{M} = \{(track_n, det_m)|n \in N, m \in M\}$
1: Initialize candidate matrix $C \leftarrow \emptyset$
2: **for** $track \in \mathcal{T}$ **do**
3:     Initialize flag matrix $F \leftarrow \emptyset$
4:     **for** $det \in \mathcal{D}$ **do**
5:         **if** $det \in track$ **then**
6:             $F \leftarrow True$
7:         **else**
8:             $F \leftarrow False$
9:     **end for**
10:    $C \leftarrow \{f \in F | f = True\}$
11: **end for**
12: Calculate the Euclidean distance $E$ of appearance
       vectors between trajectories and candidates
13: Matching using Hungarian algorithm with $E$, get $\mathcal{M}$

---

3.3.3. Motion-Mutation Filter Module

In UAV views, sudden changes in UAV motion can cause significant changes in the position of objects in adjacent frames. As the state of the objects changes dramatically under such extreme motion, the use of a solely SMF module is insufficient. Hence, we propose a MMF module to address the challenges posed by abrupt motion in UAV tracking.

Based on the UAV flight, tracing scenarios can be classified into normal and anomaly modes. In the normal mode, the UAV moves placidly, and the coupling motion between the object in the image and the UAV remains in a stable state. We calculate the matching quantity using the AMC module. When the matching quantity falls below a certain threshold $P$, it is considered to be entering the abnormal mode, in which the drone performs fast movements with different amplitudes, resulting in significant variations in the position and speed of the object in the video sequence. Luckily, the SMF's characteristic of describing states as sets allows us to address this challenge by dynamically changing the range of sets. Thus, we adopt a state set match cascade to enlarge the state space at each level to cope with the rapid drone movements at different scales. Specifically, for the predicted trajectory state $\hat{\mathbf{x}}$ through SMF, there exists a $\Delta G_k$ at level $k$, such that the trajectory state set is as shown below.

$$\hat{\mathbf{x}} = \{(\mathbf{G} + \mathbf{\Delta G})\xi + \mathbf{c} : \|\xi\|_\infty \leq 1\}. \tag{7}$$

At each level, the AMC module performs a matching process between the trajectory and object, and the level with the best performance is selected as the matching result for the current frame. An example of the MMF on the position space of objects is illustrated in Figure 2, where the red box 1 represents the trajectory to be matched and the blue box 1 represents the ground truth for trajectory 1. The $(x, y)$ and the $(x', y')$ are the position of the trajectory and target in the image, respectively. Due to the scale variance of the UAV motion, the objects that should match the trajectory are at level 2 of position, requiring a two-level state space expansion for correct matching. Similar changes in the size and velocity space also occur, where the fast movement of the drone coupled with the object motion causes changes in the velocity space, and slight variations in object size due to the movement of UAV and angle of capture result in changes in the size space.

4. Experiments

*4.1. Datasets and Metrics*

We evaluate the proposed SMFMOT model using the official MOT toolkit of VisDrone and conduct various experiments on the Visdrone2019-MOT dataset, which includes train-

ing, test, and verification sets. Each target in the dataset is annotated with a category, a tracking ID, and a bounding box. There are ten categories in the VisDrone2019-MOT dataset, namely, pedestrian, person, car, van, bus, truck, motor, bicycle, awning-tricycle, and tricycle. During the model training and evaluation process, we consider five categories, i.e., bus, car, pedestrian, van, and truck. For evaluation, we use multiple MOT metrics, such as multiple object tracking accuracy (MOTA), multiple object tracking precision (MOTP), ID switching (IDs), Identification F-Score (IDF1), and other metrics, to compare our model with other state-of-the-art methods.

### 4.2. Implementation Details

The proposed SMFMOT model is implemented in PyTorch and all experiments are conducted on a single NVIDIA GeForce RTX 3070Ti GPU. The model parameters are initialized by pre-training the model on the COCO dataset [26]. We apply random scaling, zooming, color transformation, and left–right inversion as data augmentation. The initial learning rate is set to $10^{-2}$, and the network is trained for 30 epochs in total with the SGD [27] optimizer for the detection branch network and 50 epochs for the Re-ID branch. The learning rate decays 10 times at 15 epochs. The model is trained on two NVIDIA GeForce RTX 3090 GPUs with a batch size of 6, and the input size of each frame is adjusted to $1280 \times 1280$.

### 4.3. Comparison with State-of-the-Arts

We compare our proposed SMFMOT method with state-of-the-art methods on the VisDrone2019-MOT dataset, where the combined training and validation set is used to train the network, and the test set is used to test the model performance. As shown in Table 1, our SMFMOT outperforms the state-of-the-art methods in most key metrics and ranks first on the MOTA, MOTP, IDF1, MT, ML and FN. The MOTA achieves 43.6%. The MOTP achieves 76.4%. The IDF1 achieves 54.9%. These evaluation results of other methods shown in Table 1 are taken from ref. [5], where the image pixel is set to $1920 \times 1080$ for network training. In contrast, our method uses lower-resolution images as input and achieves better performance.

**Table 1.** Comparisons of MOT methods on the VisDrone2019-MOT test set. ↑ means higher is better; ↓ means lower is better. The best results for each metric are bolded.

| Method | MOTA ↑ (%) | MOTP ↑ (%) | IDF1 ↑ (%) | MT ↑ | ML ↓ | FP ↓ | FN ↓ | IDs ↓ | FM ↓ |
|---|---|---|---|---|---|---|---|---|---|
| MOTDT [23] | −0.8 | 68.5 | 21.6 | 87 | 1196 | 44,548 | 185,453 | 1437 | 3609 |
| SORT [1] | 14.0 | 73.2 | 38.0 | 506 | 545 | 80,845 | 112,954 | 3629 | 4838 |
| IOUT [22] | 28.1 | 74.7 | 38.9 | 467 | 670 | 36,158 | 126,549 | 2393 | 3829 |
| GOG [28] | 28.7 | 76.1 | 36.4 | 346 | 836 | **17,706** | 144,657 | 1387 | **2237** |
| MOTR [29] | 22.8 | 72.8 | 41.4 | 272 | 825 | 28,407 | 147,937 | **959** | 3980 |
| TrackFormer [30] | 25 | 73.9 | 30.5 | 385 | 770 | 25,856 | 141,526 | 4840 | 4855 |
| UAVMOT [5] | 36.1 | 74.2 | 51.0 | 520 | 574 | 27,983 | 115,925 | 2775 | 7396 |
| Ours | **43.6** | **76.4** | **54.9** | **656** | **469** | 32,599 | **86,654** | 2113 | 4042 |

### 4.4. Ablation Study

To validate the effectiveness of our proposed modules, we conduct a series of ablation experiments on the VisDrone2019-MOT test set using our proposed JDR model and DeepSORT as the baseline model. As shown in Table 2, our model comprises three core components, namely, the SMF module, the AMC module, and the MMF module, for which we report the results of four critical performance metrics. The baseline achieves 42.3% on MOTA, 75.3% on MOTP, and 53.9% on IDF1. When the SMF replaces the motion model, the MOTA increases to 42.5%, the MOTP increases to 76.6%, and the IDF1 decreases to 49.7%. When the SMF replaces the motion model and the AMC module is loaded into the

baseline model, the MOTA increases to 43.3%, the MOTP achieves 76.4%, and the IDF1 achieves 53.9%. Finally, by adding the SMF module, AMC module, and MMF module to the baseline model, the MOTA increases to 43.6%, the MOTP achieves 76.4%, and the IDF1 achieves 54.9%. Due to the interdependency among the modules we designed, the ablation experiments of each module below probably depend on one or two modules.

**Table 2.** Ablation study on VisDrone2019-MOT test set. ↑ means higher is better; ↓ means lower is better. The best results for each metric are bolded.

| SMF | AMC | MMF | MOTA ↑ (%) | MOTP ↑ (%) | IDF1 ↑ (%) | IDs ↓ | IDP ↑ (%) | IDR ↑ (%) |
|---|---|---|---|---|---|---|---|---|
| ✗ | ✗ | ✗ | 42.3 | 75.3 | 53.9 | 2314 | 61.7 | 47.9 |
| ✓ | ✗ | ✗ | 42.5 | **76.6** | 49.7 | 3200 | 60.0 | 42.4 |
| ✓ | ✓ | ✗ | 43.3 | 76.4 | 53.9 | 2155 | 63.1 | 47.0 |
| ✓ | ✓ | ✓ | **43.6** | 76.4 | **54.9** | **2113** | **64.1** | **48.0** |

### 4.4.1. Analysis of SMF Module

The SMF module effectively predicts the state of the trajectory and improves the tracking accuracy. To evaluate the SMF module, we use two critical indicators (Pren, FP) on the baseline model and the baseline + SMF model. As shown in Table 3, the Prcn increases from 77.9% to 81.2%, and FP decreases from 36,868 to 28,614. The experimental results demonstrate that SMF has a positive impact on the matching stage, enabling accurate prediction of the object motion state, reducing the number of tracking errors and improving the tracking accuracy.

**Table 3.** Ablation experiments is conducted on the VisDrone2019-MOT test set using Prcn and FP as the indicators to demonstrate the effectiveness of the SMF module. ↑ means higher is better; ↓ means lower is better. The best results for each metric are bolded.

| | Prcn ↑ (%) | FP ↓ |
|---|---|---|
| Baseline | 77.9 | 36,868 |
| Baseline + SMF | **81.2** | **28,614** |

### 4.4.2. Analysis of AMC Module

The AMC module is utilized to filter out objects that do not meet the trajectory state and prevent mismatches between similar objects. To evaluate the AMC module, we use four ID association indicators (IDF1, IDS, IDP, IDR) on the baseline + SMF model and the baseline + SMF + AMC model. As presented in Table 4, the IDs decrease from 3200 to 2155. The IDF1, IDP and IDR increase from 49.7%, 60.0% and 42.4% to 53.9%, 63.1% and 47.0% (Table 3), severally. The experimental results demonstrate that AMC has a significant impact on the matching stage, and the primary contribution stems from a considerable reduction in ID switches.

**Table 4.** Ablation experiments is conducted on the VisDrone2019-MOT test set using IDF1, IDS, IDP and IDR as the indicators to demonstrate the effectiveness of the AMC module. ↑ means higher is better; ↓ means lower is better. The best results for each metric are bolded.

| | IDF1 ↑ (%) | IDs ↓ | IDP ↑ (%) | IDR ↑ (%) |
|---|---|---|---|---|
| Baseline + SMF | 49.7 | 3200 | 60.0 | 42.4 |
| Baseline + SMF + AMC | **53.9** | **2155** | **63.1** | **47.0** |

### 4.4.3. Analysis of MMF Module

The MMF module can automatically switch tracking modes based on the UAV motion state. We evaluate the MMF module based on four list ID association indicators (IDF1, IDs,

IDP, IDR) on the baseline + SMF + AMC model and baseline + SMF + AMC + MMF model. As illustrated, the IDs decrease from 2155 to 2113. The IDF1, IDP and IDR increase from 53.9%, 63.1% and 47.0% to 54.9%, 64.1% and 48.0%, respectively (Table 5). The experimental results demonstrate that MMF has a significant impact on the matching stage, and in this instance, it alleviated the influence of ID switching caused by mutation motion of UAV.

**Table 5.** Ablation experiments is conducted on the VisDrone2019-MOT test set using IDF1, IDS, IDP and IDR as the indicators to demonstrate the effectiveness of the MMF module. ↑ means higher is better; ↓ means lower is better. The best results for each metric are bolded.

|                         | IDF1 ↑ (%) | IDs ↓ | IDP ↑ (%) | IDR ↑ (%) |
|-------------------------|------------|-------|-----------|-----------|
| Baseline + SMF + AMC       | 53.9       | 2155  | 63.1      | 47.0      |
| Baseline + SMF + AMC + MMF | **54.9**   | **2113** | **64.1** | **48.0**  |

*4.5. Qualitative Results*

In this section, we showcase the implementation effectiveness of SMFMOT on UAV video sequences from the Visdrone-MOT2019 dataset more intuitively. As depicted in Figure 4, the UAV exhibits normal motion from farm10 to farm20, while it undergoes rapid motion from farm20 to farm30. Our model can effectively track multiple objects in UAV environments with normal and fast movements. The visualizations demonstrate that our model can accomplish most MOT assignments in the UAV domain.

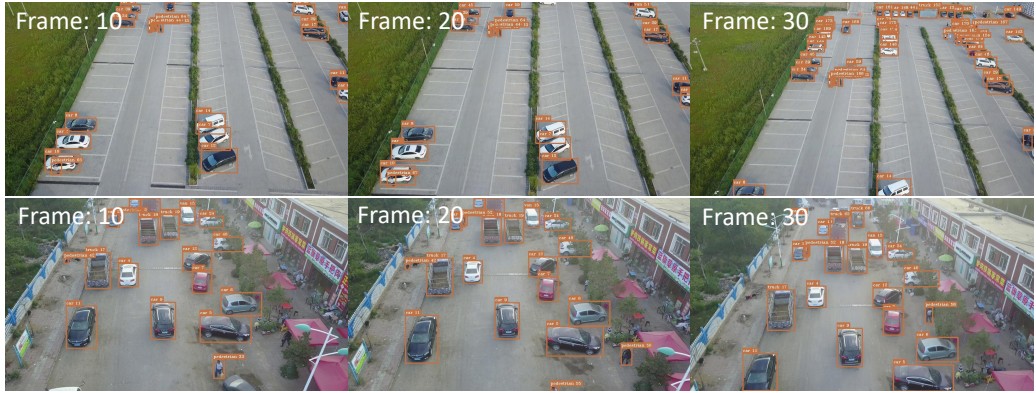

**Figure 4.** Visualization of qualitative results of the implementation performed by SMFMOT on VisDrone2019-MOT.

**5. Conclusions**

In this work, we propose a novel SMFMOT model for MOT mission in UAV, which follows the detection-based-tacking paradigm. Our model utilizes a symmetric anchor-based JDR model to accomplish both detection and Re-Identification tasks, achieving high-performance detection and high-quality Re-ID features while balancing the detection and Re-ID tasks effectively. We utilize the SMF to predict the set of trajectory states, which accurately describe the range of trajectory existence. The AMC module is proposed to optimize the appearance-based matching strategy using object status, eliminating objects that are not in the trajectory state set and reducing the effects of appearance similarity. Additionally, we propose an MMF module to address the sudden movement issue of UAV, which determines matching strategies based on the UAV motion state. The experimental results show that our proposed approach achieves state-of-the-art results for MOT, demonstrating that our method can successfully accomplish MOT mission in both normal scenarios and extreme scenarios caused by abrupt movements of UAVs. However, this work does not take into account extreme scenarios caused by natural conditions such as nighttime, storms and blizzards. Further research is required to develop a safe and reliable all-round solution that can address these challenges.

**Author Contributions:** Conceptualization, X.Y. and X.W.; methodology, X.L., R.Z. and X.Y.; software, X.Y. and R.Z.; validation, X.Y.; formal analysis, X.L. and X.Y.; investigation, X.Y. and R.Z.; resources, X.Y. and X.W.; data curation, X.Y. and R.Z.; writing—original draft preparation, R.Z.; writing—review and editing, X.L., X.Y. and R.Z.; visualization, R.Z.; supervision, X.Y. and X.W.; project administration, X.Y. and X.W.; funding acquisition, X.L. All authors have read and agreed to the published version of the manuscript.

**Funding:** This research was funded by National Natural Science Foundation of China grant number 52001029. This research was supported by the National Natural Science Foundation of China grant number 61973309 and the Natural Science Foundation of the Hunan Province, grant number 2021JJ20054.

**Data Availability Statement:** The data presented in this study are available on request from the corresponding author.

**Acknowledgments:** The authors would like to thank the Editor, and anonymous reviewers for their valuable reviews, which greatly improves the quality of this manuscript.

**Conflicts of Interest:** The authors declare no conflict of interest. The funders had no role in the design of this study; in the collection, analyses, or interpretation of data; in the writing of the manuscript; or in the decision to publish the results.

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
