# Peer review of "High-Performance Detection-Based Tracker for Multiple Object Tracking in UAVs"

_drones, doi:10.3390/drones7110681_

Round 1

Reviewer 1 Report

Comments and Suggestions for Authors

Overall, this paper presents a promising Multi-Object Tracking framework for UAV tracking, aimed at handling tiny objects with irregular motion. The proposed method has achieved impressive results on VisDrone2019-MOT.

However, there are a few points that could be further discussed in this paper: 

(1) This proposed solution may connect to a previous work [1], which also selected candidates under a spatial constraint for Re-ID matching, with the aim of dealing with irregular movements of tiny moving objects. Differences and improvements can be discussed in the section on related work. For example, the proposed method could be more efficient than [1] since it utilizes the Joint Detection and Re-ID learning (JDR) network.

(2) It might be difficult to find other drone MOT datasets than VisDrone2019-MOT, but it might be useful to compare the proposed method with more general MOT datasets to prove its robustness and generalizability. For example, previous work analyzed the irregular movements of tiny players with irregular movements in the SoccerNet dataset [2].

[1] "Tackling multiple object tracking with complicated motions—re-designing the integration of motion and appearance" (IVC 2022)
[2]"Soccernet-tracking: Multiple object tracking dataset and benchmark in soccer videos." (CVPR 2022)

Comments on the Quality of English Language

English errors are listed as follows:

Line 4: "Recently, the popular 3 multitask learning methods has been dominating..." -> "have been"
Line 15: "state-of-the-arts" -> "state-of-the-art"
Line 41: "Since Dense anchors box improve" -> "Since Dense anchors box improves"
Line 45: "Therefore, we learns Re-ID" -> "Therefore, we learn Re-ID"
Line 60: "The experimental result demonstrate" ->"The experimental results demonstrate"

Reviewer 2 Report

Comments and Suggestions for Authors

1) Summary: The paper addresses a significant problem of traffic tracking of multiple objects while the observing UAV may move. An improved trajectory prediction is proposed and this technique (SMFMOT) could be applied to other applications fields, e.g. the forestry harvester SLAM. 

2) General comments: Mention in introduction that some extreme weather and lighting conditions have not been covered. Repeat in conclusions, that the extent of results to extreme (and not necessarily very rare conditions, like nighttime, fog, storms, snow) conditions is a subject of further study in order to achieve a feasible and reliable citywide solution. 

3). Specific comments:

- Abstract: detection-based-tacking --> detection-based-tracking

- Abstract: mention that the focus is in traffic monitoring. An example: Traffic monitoring from Unmanned Aerial Vehicles (UAVs) using either preprogrammed flight paths or user-controlled behavior is an attractive solution to many urban areas. Multi-Object Tracking (MOT) is a key technology in implementing this. ...  

- Introduce W and H on P5L155 (height and width of the video frame)

- P6 Eq (2): Set x represents the range of each trajectory in the eight-dimensional space, and it can be spanned by a matrix G columns and centered by c: 

(Delete the next line).

- Results: Yu might be ebale to squeeze tables 2 and 1 to the standard width by the following changes: 

   . omit Datasets column (mention it in the text) (table 1)

   . add line change in "MOTA \\ uparrow (%)" in MOTA/MOTP/IDF1 headers (tables 1 and 2) (and IDP / IDR (table 2)

   . omit baseline column (table 2) (I think this will do it, at least try it out)

Conclusions: add a mention about future research (extensibility to difficult environmental conditions). 

Finally: congratulations to authors for good and convincing research!

Comments on the Quality of English Language

There are indeed grammatical mistakes in the text. Those will be weeded out by the editoria stuff , I guess .
